Terrien, a metabolite made by Aspergillus terreus, has activity against Cryptococcus neoformans

Cadelis Melissa 1 2 m.cadelis@auckland.ac.nz
Grey Alex 2
van de Pas Shara 2
Geese Soeren 2
http://orcid.org/0000-0003-2580-0701 Weir Bevan S. 3
Copp Brent 1
http://orcid.org/0000-0002-0467-0015 Wiles Siouxsie 2 s.wiles@auckland.ac.nz
1 School of Chemical Sciences, University of Auckland , Auckland , New Zealand
2 Bioluminescent Superbugs Lab, Department of Molecular Medicine and Pathology, University of Auckland , Auckland , New Zealand
3 Manaaki Whenua – Landcare Research , Auckland , New Zealand
Krajaejun Theerapong
Electronic publication date: 2022 Oct 18
Publication date: 2022
Volume: 10
Electronic Location ID: e14239
Received 2022 Jun 29; Accepted 2022 Sep 23
Copyright: © 2022 Cadelis et al.
Copyright year: 2022
Copyright holder: Cadelis et al.
License: This is an open access article distributed under the terms of the Creative Commons Attribution License, which permits unrestricted use, distribution, reproduction and adaptation in any medium and for any purpose provided that it is properly attributed. For attribution, the original author(s), title, publication source (PeerJ) and either DOI or URL of the article must be cited.
License URL: https://creativecommons.org/licenses/by/4.0/

Keywords: Antibiotic discovery, Natural compounds, Anti-fungal activity, Antimicrobial testing, Fungal secondary metabolites, Mycobacteria, Staphylococcus aureus, Cryptococcus neoformans

Funding: Cure Kids New Zealand Carbon Farming This work was supported by Cure Kids and New Zealand Carbon Farming. The funders had no role in study design, data collection and analysis, decision to publish, or preparation of the manuscript.

==============================
Antimicrobial compounds, including antibiotics, have been a cornerstone of modern medicine being able to both treat infections and prevent infections in at-risk people, including those who are immune-compromised and those undergoing routine surgical procedures. Their intense use, including in people, animals, and plants, has led to an increase in the incidence of resistant bacteria and fungi, resulting in a desperate need for novel antimicrobial compounds with new mechanisms of action. Many antimicrobial compounds in current use originate from microbial sources, such as penicillin from the fungus Penicillium chrysogenum (renamed by some as P. rubens). Through a collaboration with Aotearoa New Zealand Crown Research Institute Manaaki Whenua–Landcare Research we have access to a collection of thousands of fungal cultures known as the International Collection of Microorganisms from Plants (ICMP). The ICMP contains both known and novel species which have not been extensively tested for their antimicrobial activity. Initial screening of ICMP isolates for activity against Escherichia coli and Staphylococcus aureus directed our interest towards ICMP 477, an isolate of the soil-inhabiting fungus, Aspergillus terreus. In our investigation of the secondary metabolites of A. terreus, through extraction, fractionation, and purification, we isolated nine known natural products. We evaluated the biological activity of selected compounds against various bacteria and fungi and discovered that terrein (1) has potent activity against the important human pathogen Cryptococcus neoformans.

Introduction

Antibiotics are a cornerstone of modern medicine, used to treat infectious diseases and prevent infection in vulnerable patients, such as those undergoing surgery, or treatment for cancer. A 2014 report by the World Health Organization concluded that far from being an apocalyptic fantasy, resistance to antibiotics means the post-antibiotic era is a very real possibility for the 21st century (World Health Organization, 2014) and will make childbirth, routine surgery, organ transplantation, and cancer chemotherapy life-threateningly risky. A recent comprehensive assessment of the global burden of antimicrobial resistance estimated that globally in 2019 there were almost 5 million deaths associated with bacterial resistance to antibiotics, including 1.27 million deaths that were directly attributable to antibiotic resistance (Antimicrobial Resistance Collaborators, 2022). The six leading pathogens were Escherichia coli, Staphylococcus aureus, Klebsiella pneumoniae, Streptococcus pneumoniae, Acinetobacter baumannii, and Pseudomonas aeruginosa. Together these bacteria were responsible for 3.57 million of the deaths associated with resistance, and 929,000 of the deaths directly attributable to resistance (Antimicrobial Resistance Collaborators, 2022).

One of the keys to managing this crisis is to develop new antibiotics which necessitates the identification of new chemical compounds with novel modes of action. During the golden age of antibiotic discovery, fungi were a significant source of antimicrobial activity responsible for several important classes of antibiotic still in use today, including penicillin from the fungus Penicillium chrysogenum (renamed by some as P. rubens). Despite previous assumptions that fungi have largely been a tapped resource, the search for antimicrobial compounds from fungi has not been thorough. This is evidenced by the wealth of novel compounds discovered in recent years (Cheng et al., 2012; Ishii et al., 2013; Wu et al., 2015; Myrtle, Beekman & Barrow, 2016). Even a re-examination of previously screened genera such as Penicillium has revealed an enormous wealth of secondary metabolites that may have been overlooked (Nielsen et al., 2017).

Through the Crown Research Institute Manaaki Whenua, we have access to fungal isolates from the International Collection of Microorganisms from Plants (ICMP), a collection of cultures of fungi and bacteria primarily sourced from Aotearoa New Zealand and the South Pacific (Johnston, Weir & Cooper, 2017). The collection contains many new species and even new genera. Due to a longstanding interest in drug discovery for tuberculosis (Dalton et al., 2016; Dalton et al., 2017), we have previously begun screening the collection for anti-mycobacterial activity and have identified several ICMP isolates including an unknown species of Boeremia and an isolate of an unknown genus and species in the family Phanerochaetaceae (Grey et al., 2021). More recently, we have expanded our efforts to screen for activity against three of the important human pathogens described earlier, E. coli, K. pneumoniae, S. aureus.

We have previously reported that ICMP 477, an isolate of the soil-inhabiting fungus Aspergillus terreus, has activity against Mycobacterium abscessus and M. marinum which is retained after extraction (Grey et al., 2021). Here we report our further investigations of ICMP 477 including fermentation, extraction, purification of natural products and the biological activities of these natural products. We show that terrein (1) has potent activity against the opportunistic human pathogen Cryptococcus neoformans.

Materials and Methods

General experimental procedures

Mass spectra were acquired on a Bruker micrOTOF Q II spectrometer. Melting points were recorded on an electrothermal melting point apparatus and are uncorrected. 1H and 13C NMR spectra were recorded at 298 K on a Bruker AVANCE 400 spectrometer at 400 and 100 MHz, respectively using standard pulse sequences. Proto–deutero solvent signals were used as internal references (CD3OD: δH 3.31, δC 49.0; CDCl3: δH 0.00 (TMS), δC 77.16). For 1H NMR, the data are quoted as position (δ), relative integral, multiplicity (s = singlet, d = doublet, t = triplet, q = quartet, p = pentet m = multiplet, dd = doublet of doublets, ddd = doublet of doublets of doublets, td = triplet of doublets, br = broad), coupling constant (J, Hz), and assignment to the atom. The 13C NMR data are quoted as position (δ), and assignment to the atom. Flash column chromatography was carried out using Kieselgel 60 (Merck, Darmstadt, Germany) silica gel or Diol (Merck, Darmstadt, Germany) bonded silica x (40–63 μm), C8 (Merck, Darmstadt, Germany) reversed–phase (40–63 μm) solid support. Gel filtration flash chromatography was carried out on Sephadex LH–20 (Merck, Germany). Thin layer chromatography was conducted on DC–plastikfolien Kieselgel 60 F254 plates (Merck, Darmstadt, Germany). All solvents used were of analytical grade or better and/or purified according to standard procedures.

Fungal material

As previously described, culture ICMP 477 was isolated in September 1961 in Auckland, Aotearoa New Zealand, from sheep’s wool incubated at 30 °C (Wiles & Geese, 2022). The identification of this culture as A. terreus is supported by GenBank sequence MW862777. A. terreus is a common cosmopolitan saprotrophic soil-inhabiting fungus (Samson et al., 2011), although it is occasionally recorded as a human pathogen (Lass-Flörl, 2018). Cultures of A. terreus were grown at room temperature on Potato Dextrose Agar (PDA) (Fort Richard, New Zealand) plates.

Zone of inhibition assays

Zone of inhibition testing was carried out as previously described (Wiles & Geese, 2022). Specifically, potato dextrose agar (PDA) plates were inoculated with a lawn of either antibiotic-sensitive E. coli (ATCC 25922) (In vitro Technologies, Auckland, New Zealand) or resistant clinical isolates (CTX-M-9, CTX-M-14, CTX-M-15, NDM-1 (obtained from Auckland Hospital)) or antibiotic-resistant K. pneumoniae (ATCC 700603, produces SHV-18) (In vitro Technologies, Auckland, New Zealand). Similarly, Mueller-Hinton agar plates were inoculated with a lawn of either antibiotic-sensitive S. aureus (ATCC 29213) or antibiotic-resistant S. aureus (ATCC 33593) (In vitro Technologies, Auckland, New Zealand). ICMP 477 was grown on PDA plates for 36 days and fungal plugs removed using a 6 mm punch biopsy tool (Amtech Medical, Whanganui, New Zealand). Fungal plugs were placed onto the bacterial lawns, alongside PDA plugs containing no fungus as a control. Plates were incubated inverted at 37 °C for 24 h before measuring any zones of inhibition (in mm) produced. Experiments were performed using three biological replicates with three technical replicates each.

Fermentation, extraction, and isolation

Compounds were isolated as previously described in Cadelis et al. (2020). Specifically, forty PDA plates were inoculated with ICMP 477 and incubated at room temperature for 3 weeks. Fully grown fungal plates were freeze-dried (26.57 g, dry weight) and extracted with MeOH (2 × 500 mL) for 4 h followed CH2Cl2 (2 × 500 mL) overnight. Combined organic extracts were concentrated under reduced pressure to afford an orange oil (2.45 g). The crude product was subjected to C8 reversed-phase column chromatography eluting with a gradient of H2O/MeOH to afford five fractions (F1–F5). Fraction F3 was subjected to purification by Sephadex LH-20, eluting with MeOH/5% CH2Cl2, to afford terrein (1) (3.05 mg), aspulvinone E (2) (4.22 mg) and aspulvinone G (3) (4.43 mg). Further separation of F4 by Sephadex LH-20, eluting with MeOH/5% CH2Cl2, afforded six fractions (A1–A6). Purification of fraction A2 using Diol-bonded silica gel column chromatography, eluting with gradient n-hexane/EtOAc, afforded terretonin (4) (19.60 mg) and terretonin A (5) (4.91 mg) while purification of A4 using the same method afforded asperteretal B (6) (2.39 mg), flavipesolide C (7) (1.44 mg) and butyrolactone II (8) (1.98 mg). Fraction F5 was subjected to purification by Sephadex LH-20, eluting with MeOH/5% CH2Cl2, to afford butyrolactone I (9) (36.12 mg).

Terrein (1)

1H NMR (CD3OD, 400 MHz) δ 6.86 (1H, dq, J = 15.8, 6.9 Hz, H-7), 6.46 (1H, dq, J = 15.8, 1.4 Hz, H-6), 6.03 (1H, s, H-2), 4.71 (1H, d, J = 2.4 Hz, H-4), 4.11 (1H, d, J = 2.4 Hz, H-5), 1.96 (3H, dd, J = 6.9, 1.4 Hz, H3-8); 13C NMR (CD3OD, 100 MHz) δ 205.7 (C-1), 170.9 (C-3), 141.9 (C-7), 126.4 (C-6), 124.9 (C-2), 82.4 (C-5), 78.1 (C-4), 19.5 (C-8); (+)-HRESIMS m/z 155.0705 [M+H]+ (calcd for C8H11O3, 155.0703).

Aspulvinone E (2)

m.p. 281–283 °C; 1H NMR (CD3OD, 400 MHz) δ 7.95 (2H, d, J = 8.7 Hz, H-2′), 7.66 (2H, d, J = 8.7 Hz, H-8), 6.80 (2H, d, J = 8.7 Hz, H-9), 6.77 (2H, d, J = 8.7 Hz, H-3′), 6.22 (1H, s, H-6); 13C NMR (CD3OD, 100 MHz) δ 179.3 (C-4), 171.0 (C-2), 158.3 (C-10), 155.0 (C-4′), 147.8 (C-5), 132.6 (C-8), 128.2 (C-2′), 127.5 (C-7), 127.1 (C-1′), 116.4 (C-9), 115.5 (C-3′), 103.8 (C-6), 94.2 (C-3); (+)-HRESIMS m/z 319.0578 [M+Na]+ (calcd for C17H12NaO5, 319.0577).

Aspulvinone G (3)

m.p. 264–265 °C; 1H NMR (CD3OD, 400 MHz) δ 7.76 (1H, d, J = 8.5 Hz, H-6′), 7.65 (2H, d, J = 8.8 Hz, H-8), 6.81 (2H, d, J = 8.8 Hz, H-9), 6.33 (1H, dd, J = 8.5, 2.5 Hz, H-5′), 6.29 (1H, d, J = 2.5 Hz, H-3′), 6.18 (1H, s, H-6); 13C NMR (CD3OD, 100 MHz) δ 176.5 (C-4), 170.8 (C-2), 158.7 (C-10, C-2′), 157.7 (C-4′), 146.0 (C-5), 132.7 (C-8), 129.2 (C-6′), 127.1 (C-7), 116.5 (C-9), 113.9 (C-1′), 107.7 (C-5′), 105.4 (C-3′), 104.7 (C-6), 96.3 (C-3); (+)-HRESIMS m/z 335.0536 [M+Na]+ (calcd for C17H12NaO6, 335.0526).

Terretonin (4)

m.p. 288–290 °C [lit m.p. 289–291 °C (Li et al., 2019)]; [α]D21 = −76.2 (c = 0.10, MeOH) [lit [α]D25 = −73.7 (c = 0.19, MeOH) (Li et al., 2019)]; 1H NMR (CDCl3, 400 MHz) δ 6.15 (1H, s, 6-OH), 5.48 (1H, s, H2-17A), 5.09 (1H, s, H2-17B), 3.80 (3H, s, OMe), 3.54 (1H, s, H-4a), 2.98 (1H, d, J = 14.4 Hz, H2-11A), 2.75–2.68 (1H, m, H2-9A), 2.56–2.48 (1H, m, H2-9B), 2.41–2.33 (1H, m, H2-10A), 2.26 (1H, d, J = 14.4 Hz, H2-11B), 1.93 (3H, s, H3-13), 1.82–1.74 (1H, m, H2-10B), 1.72 (3H, s, H3-19), 1.48 (6H, s, H3-14, H3-15), 1.45 (3H, s, H3-18), 1.22 (3H, s, H3-16); 13C NMR (CDCl3, 100 MHz) δ 214.1 (C-8), 201.6 (C-1), 197.1 (C-5), 168.7 (C-20), 167.9 (C-4), 140.0 (C-12), 138.8 (C-6), 131.7 (C-6a), 117.3 (C-17), 85.7 (C-2), 77.7 (C-10b), 53.9 (OMe), 52.5 (C-4b), 49.6 (C-12a), 48.0 (C-7), 44.7 (C-4a), 43.3 (C-10a), 35.0 (C-11), 32.8 (C-9), 28.3 (C-10), 23.7 (C-18), 23.6 (C-14), 21.4 (C-15, C-19), 20.0 (C-13), 18.7 (C-16); (+)-HRESIMS m/z 511.1949 [M+Na]+ (calcd for C26H32NaO9, 511.1940).

Terretonin A (5)

m.p. 231–233 °C [lit m.p. 232–233 °C (Li et al., 2005)]; [α]D21 = −60 (c = 0.125, CHCl3) [lit [α]D20 = −115.7 (c = 0.14, CHCl3) (Li et al., 2005)]; 1H NMR (CDCl3, 400 MHz) δ 6.18 (1H, s, 6-OH), 5.18 (1H, s, H2-17A), 4.98 (1H, s, H2-17B), 3.83 (3H, s, OMe), 2.84 (1H, s, H-4a), 2.69 (1H, dd, J = 19.0, 9.0 Hz, H2-9A), 2.58–2.48 (2H, m, H2-9B, H2-11A), 2.33 (1H, dd, J = 13.8, 2.5 Hz, H2-11B), 2.19 (1H, dd, J = 13.8, 9.0 Hz, H2-10A), 1.83 (3H, s, H3-13), 1.74 (3H, s, H3-19), 1.70 (1H, dd, J = 13.8, 10.0 Hz, H2-10B), 1.56–1.52 (1H, m, H-10b), 1.47 (3H, s, H3-14), 1.46 (3H, s, H3-15), 1.44 (3H, s, H3-18), 1.11 (3H, s, H3-16); 13C NMR (CDCl3, 100 MHz) δ 214.0 (C-8), 201.4 (C-1), 198.1 (C-5), 168.6 (C-20), 167.1 (C-4), 143.2 (C-12), 139.2 (C-6), 137.4 (C-6a), 112.3 (C-17), 86.0 (C-2), 53.9 (OMe), 53.1 (C-10b), 50.7 (C-12a), 49.3 (C-4a), 48.1 (C-7), 46.0 (C-4b), 38.4 (C-10a), 34.5 (C-10), 32.8 (C-9), 29.3 (C-11), 23.9 (C-18), 23.7 (C-14), 22.2 (C-19), 21.1 (C-15), 17.0 (C-16), 16.7 (C-13); (+)-HRESIMS m/z 495.1993 [M+Na]+ (calcd for C26H32NaO8, 495.1990).

Asperteretal B (6)

[α]D21 = −10.9 (c = 0.10, MeOH) [lit [α]D20 = −18.6 (c = 0.12, MeOH) (Guo et al., 2016)]; 1H NMR (CD3OD, 400 MHz) δ 7.52 (2H, d, J = 8.8 Hz, H-2′), 6.90 (1H, dd, J = 8.3, 2.6 Hz, H-8), 6.80 (2H, d, J = 8.8 Hz, H-3′), 6.69 (1H, d, J = 2.6 Hz, H-12), 6.68 (1H, d, J = 8.3 Hz, H-9), 5.31–5.27 (1H, m, H-14), 3.25 (2H, d, J = 7.1 Hz, H2-13), 1.72 (3H, s, H3-16), 1.68 (3H, s, H3-17); 13C NMR (CD3OD, 100 MHz) δ 174.0 (C-2), 166.0 (COOH), 160.2 (C-4′), 156.8 (C-4), 154.0 (C-10), 132.4 (C-15), 131.4 (C-2′), 129.8 (C-12), 128.8 (C-11), 128.0 (C-3), 126.8 (C-8), 126.3 (C-7), 123.6 (C-14), 121.9 (C-1′), 115.9 (C-3′), 115.5 (C-9), 102.8 (C-5), 29.8 (C-6), 28.9 (C-13), 25.7 (C-16), 17.5 (C-17); (–)-HRESIMS m/z 409.1287 [M-H]– (calcd for C23H21O7, 409.1293).

Flavipesolide C (7)

1H NMR (CD3OD, 400 MHz) δ 7.53 (2H, d, J = 8.9 Hz, H-2′), 6.97–6.92 (1H, m, H-8), 6.92–6.89 (1H, m, H-12), 6.81 (2H, d, J = 8.9 Hz, H-3′), 6.64 (1H, d, J = 8.9 Hz, H-9), 3.82–3.77 (2H, m, H2-13), 2.74 (2H, t, J = 6.3 Hz, H2-13), 1.80 (2H, t, J = 6.3 Hz, H2-14), 1.26 (6H, s, H3-16, H3-17); 13C NMR (CD3OD, 100 MHz) δ 174.1 (C-2), 166.0 (COOH), 160.7 (C-4′), 156.9 (C-4), 153.6 (C-10), 131.5 (C-2′), 129.2 (C-12), 128.2 (C-3), 127.8 (C-8), 126.3 (C-7), 122.0 (C-11), 121.9 (C-1′), 118.1 (C-9), 116.4 (C-3′), 103.0 (C-5), 75.4 (C-15), 33.8 (C-14), 30.1 (C-6), 27.0 (C-16, C-17), 23.5 (C-13); (–)-HRESIMS m/z 409.1284 [M-H]– (calcd for C23H21O7, 409.1293).

Butyrolactone II (8)

[α]D23 = +6.2 (c = 0.13, acetone) [lit [α]D29 = +4.8 (c = 0.45, acetone) (Dewi, Tachibana & Darmawan, 2014)]; 1H NMR (CD3OD, 400 MHz) δ 7.61 (2H, d, J = 8.9 Hz, H-2′), 6.89 (2H, d, J = 8.9 Hz, H-3′), 6.67 (2H, d, J = 8.5 Hz, H-8), 6.54 (2H, d, J = 8.5 Hz, H-9), 3.81 (3H, s, OMe), 3.49 (2H, s, H2-6); 13C NMR (CD3OD, 100 MHz) δ 170.4 (COOMe), 169.8 (C-2), 157.7 (C-4′), 156.3 (C-10), 137.7 (C-3), 132.4 (C-8), 130.5 (C-2′), 129.2 (C-4), 123.7 (C-7), 120.9 (C-1′), 116.5 (C-3′), 115.5 (C-9), 85.2 (C-5), 53.4 (OMe), 39.3 (C-6); (–)-HRESIMS m/z 355.0819 [M-H]– (calcd for C19H15O7, 355.0823).

Butyrolactone I (9)

[α]D21 = +46.7 (c = 0.25, MeOH) [lit [α]D23 = +68.3 (c = 0.3, MeOH) (Dewi, Tachibana & Darmawan, 2014)]; 1H NMR (CD3OD, 400 MHz) δ 7.61 (2H, d, J = 8.8 Hz, H-2′), 6.89 (2H, d, J = 8.8 Hz, H-3′), 6.56 (1H, dd, J = 8.0, 1.9 Hz, H-8), 6.52 (1H, d, J = 8.0 Hz, H-9), 6.44 (1H, d, J = 1.9 Hz, H-12), 5.12–5.07 (1H, m, H-14), 3.81 (3H, s, OMe), 3.46 (2H, s, H2-6), 3.10 (2H, d, J = 7.7 Hz, H2-13), 1.69 (3H, s, H3-16), 1.60 (3H, s, H3-17); 13C NMR (CD3OD, 100 MHz) δ 171.7 (COOMe), 170.4 (C-2), 159.4 (C-4′), 155.1 (C-10), 139.8 (C-3), 132.4 (C-12), 130.4 (C-2′), 129.8 (C-8), 129.2 (C-11), 128.5 (C-4), 125.1 (C-7), 123.6 (C-14), 123.2 (C-1′), 116.6 (C-3′), 115.1 (C-9), 86.8 (C-5), 53.9 (OMe), 39.7 (C-6), 28.7 (C-13), 26.0 (C-16), 17.8 (C-17); (+)-HRESIMS m/z 447.1406 [M+Na]+ (calcd for C24H24NaO7, 447.1414).

Testing extracts for antimicrobial activity

Data were collected as described previously (Cadelis et al., 2021a; Wiles, 2022). Specifically, dry samples of extracts were dissolved in DMSO (Sigma-Aldrich, St. Louis, MO, USA) to make a 25 mg/mL solution and then further diluted into Mueller Hinton broth II (MHB) (Fort Richard, Auckland, New Zealand) to achieve a maximum concentration of 2 mg/mL (Andrews, 2001). Each extract (200 µL) was tested in duplicate by adding to two adjacent wells along the top of the 96-well plate (Thermo Fisher, NUN167008, Waltham, MA, USA). MHB (100 µL) was then added to the remaining wells and extract solution (100 µL) serially diluted two-fold down the plate and discarded. Aliquots of bioluminescent derivatives of S. aureus (Xen 36 (PerkinElmer Inc., MA, USA)) and E. coli (25922 lux (Robertson, Gizdavic-Nikolaidis & Swift, 2018)), at an optical density at 600 nm of 0.01 (approximately 1 × 106 colony forming units (CFU)/mL) were then added to all the wells. This gave a maximum concentration of 1 mg/mL and a minimum concentration of 16 µg/mL. The maximum volume/volume concentration of DMSO in all extracts was 4%; therefore, the negative control was tested at an identical concentration.

Luminescence was measured using a Victor X-3 luminescence plate reader (PerkinElmer Inc., MA, USA) with an integration time of 1s at 0, 2, 4 and 24 h to determine the minimum inhibitory concentration (MIC), between which times the plates were incubated at 37 °C with shaking at 100 rpm. MIC was defined as causing a 1 log reduction in bacterial light production. After 24 h, 10 µL of liquid from all wells showing inhibition of bacterial growth was pipetted onto a plate of MH agar. Once all liquid had evaporated, the plates were then incubated inverted at 37 °C for 16–20 h, and the minimum bactericidal concentration (MBC) was measured (Andrews, 2001).

Testing pure compounds for antimicrobial activity

Antimicrobial evaluation of the pure compounds against Acinetobacter baumannii ATCC 19606, Candida albicans ATCC 90028, Cryptococcus neoformans ATCC 208821, E. coli ATCC 25922, Klebsiella pneumoniae ATCC 700603, Pseudomonas aeruginosa ATCC 27853, and S. aureus ATCC 43300 (MRSA) was undertaken at the Community for Open Antimicrobial Drug Discovery at The University of Queensland (Queensland, Australia) according to their standard protocols (Blaskovich et al., 2015).

As described previously (Blaskovich et al., 2015; Cadelis et al., 2019; Cadelis et al., 2020; Cadelis et al., 2021a; Cadelis et al., 2021b) bacterial strains were cultured in either Luria broth (LB) (In Vitro Technologies, USB75852, Victoria, Australia), nutrient broth (NB) (Becton Dickson, 234,000, New South Wales, Australia) or MHB at 37 °C overnight (Blaskovich et al., 2015). A sample of culture was then diluted 40-fold in fresh MHB and incubated at 37 °C for 1.5−2 h. The compounds were serially diluted 2-fold across the wells of 96-well plates (Corning 3641, nonbinding surface), with compound concentrations ranging from 0.015 to 64 μg/mL. The resultant mid log phase cultures were diluted to the final concentration of 1 × 106 CFU/mL. Then, 50 μL was added to each well of the compound containing plates, giving a final compound concentration range of 0.008–32 μg/mL and a cell density of 5 × 105 CFU/mL. All plates were then covered and incubated at 37 °C for 18 h. Resazurin was added at 0.001% final concentration to each well and incubated for 2 h before MICs were read by eye. Experiments were carried out in duplicate.

Fungal strains were cultured for 3 days on yeast potato dextrose (YPD) agar at 30 °C. A yeast suspension of 1 × 106 to 5 × 106 CFU/mL was prepared from five colonies. These stock suspensions were diluted with yeast nitrogen base (YNB) (233520; Becton Dickinson, New South Wales, Australia) broth to a final concentration of 2.5 × 103 CFU/mL. The compounds were serially diluted 2-fold across the wells of 96-well plates (Corning 3641, nonbinding surface), with compound concentrations ranging from 0.015 to 64 μg/mL and final volumes of 50 μL, plated in duplicate. Then, 50 μL of a previously prepared fungi suspension, in YNB broth to the final concentration of 2.5 × 103 CFU/mL, was added to each well of the compound-containing plates, giving a final compound concentration range of 0.008–32 μg/mL. Plates were covered and incubated at 35 °C for 36 h without shaking. Candida albicans MICs were determined by measuring the absorbance at an optical density (OD) of 530 nm (OD530). For Cryptococcus neoformans, resazurin was added at 0.006% final concentration to each well and incubated for a further 3 h before MICs were determined by measuring the absorbance at OD570–600. Experiments were carried out in duplicate.

Colistin and vancomycin were used as positive bacterial inhibitor standards for Gram-negative and Gram-positive bacteria, respectively. Fluconazole was used as a positive fungal inhibitor standard for Candida albicans and Cryptococcus neoformans. The antibiotics were provided in four concentrations, with two above and two below their MIC value, and plated into the first eight wells of column 23 of the 384-well NBS plates. The quality control (QC) of the assays was determined by the antimicrobial controls and the Z’-factor (using positive and negative controls). Each plate was deemed to fulfil the quality criteria (pass QC), if the Z’-factor was above 0.4, and the antimicrobial standards showed full range of activity, with full growth inhibition at their highest concentration, and no growth inhibition at their lowest concentration (Blaskovich et al., 2015).

Antimicrobial evaluation against M abscessus and M marinum was undertaken using in-house assays with the bioluminescent derivatives M. abscessus BSG301 (Cadelis et al., 2021b) and M. marinum BSG101 (Dalton et al., 2017). Assays were performed as described previously in detail on protocols.io (accessed June 2022) (Wiles, 2021a,2021b) and in Grey et al. (2021). Specifically, mycobacterial cultures were grown shaking at 200 rpm in Middlebrook 7H9 broth (Fort Richard, Auckland, New Zealand) supplemented with 10% Middlebrook ADC enrichment media (Fort Richard, Auckland, New Zealand), 0.4% glycerol (Sigma-Aldrich, St. Louis, MO, USA), and 0.05% tyloxapol (Sigma-Aldrich, St. Louis, MO, USA). M. abscessus was grown at 37 °C and M. marinum at 28 °C. Cultures were grown until they reached the stationary phase (approximately 3–5 days for M. abscessus BSG301 and 7–10 days for M. marinum BSG101) and then diluted in MHB supplemented with 10% Middlebrook ADC enrichment media and 0.05% tyloxapol to give an OD600 of 0.001, which is the equivalent of ~106 bacteria per mL. Pure compounds were dissolved in DMSO and added to the wells of a black 96-well plate (Nunc, Thermo Scientific, Waltham, MA, USA) at doubling dilutions with a maximum concentration of 128 μg/mL. Then, 50 μL of diluted bacterial culture was added to each well of the compound containing plates giving final compound concentrations of 0–64 μg/mL and a cell density of ~5 × 105 CFU/mL. Rifampicin (Sigma-Aldrich, St. Louis, MO, USA) was used as positive control at 1,000 μg/mL for M. abscessus and 10 μg/mL for M. marinum. Between measurements, plates were covered, placed in a plastic box lined with damp paper towels, and incubated with shaking at 100 rpm at 37 °C for M. abscessus and 28 °C for M. marinum. Bacterial luminescence was measured at regular intervals over 72 h using a Victor X-3 luminescence plate reader with an integration time of 1 s. We defined the MIC as causing a 1-log reduction in light production, as previously described (Dalton et al., 2016). Experiments were carried out in triplicate and repeated if there was sufficient compound.

Statistical analysis

Statistical analysis of E. coli zone of inhibition data was performed using GraphPad Prism version 9.4.1. Data was analysed using a one-way ANOVA using the Kruskal-Wallis test for multiple comparisons. Statistical significance was set at p ≤ 0.05 (adjusted for multiple comparisons).

Results

A. terreus ICMP 477 was tested for production of zones of inhibition against a panel of antibiotic-sensitive and resistant isolates of E. coli, K. pneumoniae, and S. aureus (Fig. 1). None of the three biological replicates of ICMP 477 resulted in any zones of inhibition with K. pneumoniae ATCC 700603. All three biological replicates of ICMP 477 produced zones against S. aureus, with no difference between the antibiotic-sensitive isolate (ATCC 29213, median zone of 10 mm) and antibiotic-resistant isolate (ATCC 33593, median zone of 9 mm). Interestingly, only two of the three biological replicates of ICMP 477 produced zones of inhibition against E. coli. Of these active cultures, the zones of inhibition produced for the antibiotic-sensitive isolate (ATCC 25922, median zone of 15 mm) were significantly smaller than for the resistant clinical isolates CTX-M-15 (median zone of 20 mm; p = 0.0027, Kruskal-Wallis test) and NDM-1 (median zone of 21 mm; p = 0.0002, Kruskal-Wallis test).

Figure 1 Activity of ICMP 477 against antibiotic-sensitive and resistant strains of E. coli, K. pneumoniae, and S. aureus.

Key: E. coli ATCC 25922, antibiotic-sensitive; E. coli CTX-M-9, CTX-M-14, CTX-M-15, and NDM-1, resistant clinical isolates; K. pneumoniae ATCC 700603, antibiotic-resistant produces SHV-18; S. aureus ATCC 29213, antibiotic-sensitive; S. aureus ATCC 33593, methicillin-resistant. Data are presented as dot plots of the zones of inhibition (in mm) after 24 h incubation with the fungus. Each replicate experiment is shown in a different shade of blue. The dotted line denotes the size of the fungal plugs and is used to indicate the no inhibition control. Raw data is available at: https://doi.org/10.17608/k6.auckland.20760364.v1.

Freeze-dried PDA plates (26.57 g, dry weight) inoculated with A. terreus ICMP 477 were extracted with a combination of methanol and dichloromethane to afford a crude extract. The crude extract was subjected to C8 reversed-phase column chromatography (H2O/MeOH) which afforded five fractions. The crude extract and five fractions were tested for antimicrobial activity against E. coli and S. aureus. Neither the crude extract or the five fractions showed any significant activity against E. coli (Fig. 2) while fractions 4 and 5 showed some activity against S. aureus at 1 mg/mL and 125 μg/mL, respectively (Fig. 3).

Figure 2 Activity of the crude extract and fractions from ICMP 477 against E. coli 25922 lux.

Key: Fraction 1, 100% water; fraction 2, 25% MeOH; fraction 3, 50% MeOH; fraction 4, 75% MeOH; fraction 5, 100% MeOH. Data are presented as box and whisker plots of the log reduction in area under curve (AUC) values derived from the luminescence output over 24 h when compared to the no extract/fraction DMSO control. Boxes are upper and lower quartiles with median shown. The whiskers extend up to 1.5× the inter-quartile range activity scores. The dotted line denotes a 90% reduction in AUC compared to the no extract/fraction DMSO control and is used to denote the minimum inhibitory concentration. Raw data is available at: https://doi.org/10.17608/k6.auckland.20113922.v1.

Figure 3 Activity of the crude extract and fractions from ICMP 477 against S. aureus Xen36.

Key: Fraction 1, 100% water; fraction 2, 25% MeOH; fraction 3, 50% MeOH; fraction 4, 75% MeOH; fraction 5, 100% MeOH. Data are presented as box and whisker plots of the log reduction in area under curve (AUC) values derived from the luminescence output over 24 h when compared to the no extract/fraction DMSO control. Boxes are upper and lower quartiles with median shown. The whiskers extend up to 1.5× the inter-quartile range activity scores. The dotted line denotes a 90% reduction in AUC compared to the no extract/fraction DMSO control and is used to denote the minimum inhibitory concentration. Raw data is available at: https://doi.org/10.17608/k6.auckland.20113922.v1.

Analysis of fractions 1–5 by NMR spectroscopy showed that fractions 1 and 2 were comprised of primary metabolites, fractions 3 and 4 appeared to contain secondary metabolites, while fraction 5 contained predominantly fatty acids and sterols. Thus, fractions 3–5 were subjected to extensive chromatographic methods for purification including Sephadex LH-20 (MeOH/5% CH2Cl2) and Diol-bonded silica gel (hexane/EtOAc) chromatography to afford compounds 1–9 (Fig. 4). Structure elucidation of compounds 1–9 was achieved by a combination of NMR spectroscopy and mass spectrometry, which agreed with literature data. Compounds 1–9 were identified as terrein (1) (Raistrick & Smith, 1935), aspulvinone E (2) (Ojima, Takenaka & Seto, 1973), aspulvinone G (3) (Ojima, Takenaka & Seto, 1975), terretonin (4) (Springer et al., 1979), terretonin A (5) (Li et al., 2005), asperteretal B (6) (Guo et al., 2016), flavipesolide C (7) (Wang et al., 2016), butyrolactone II (8) (Rao et al., 2000) and butyrolactone I (9) (Rao et al., 2000).

Figure 4 Structures of isolated natural products 1–9.

We prioritized testing samples for antimicrobial activity against a panel of important human pathogens: A. baumannii, Candida albicans, Cryptococcus neoformans, E. coli, K. pneumoniae, P. aeruginosa and methicillin-resistant S. aureus (MRSA) (Table 1). These experiments were carried out in duplicate. Compounds 6–8 were not tested in these assays as we did not have enough of each sample. Of the tested compounds, one compound, Terrein (1), exhibited potent antifungal activity against C. neoformans. None of the tested compounds exhibited significant activity against any of the bacterial strains.

Table 1 Antimicrobial and antifungal activities of natural products 1–5 and 9.

Compound	Percentage inhibition at 32 µg/mL	
S. a a	E. c b	K. p c	P. a d	A. b e	C. a f	C. n g	
1	12.61	19.64	18.29	−5.74	27.04	6.09	105.48	
2	0.46	17.02	0	10.26	−8.91	4.01	27.28	
3	58.18	17.38	−7.4	22.83	15.59	8.88	−5.27	
4	11.51	−1.55	9.45	9.09	5.76	5.7	3.75	
5	14.46	−0.77	11.04	12.14	2.59	4.21	1.31	
9	1.18	14.74	27.33	10.01	30.58	16.75	−33.38	
Notes:

All values are presented as the mean (n = 2).

a Staphylococcus aureus ATCC 43300 (MRSA) with vancomycin (MIC 1 μg/mL) used as a positive control.

b Escherichia coli ATCC 25922 with colistin (MIC 0.125 μg/mL).

c Klebsiella pneumoniae ATCC 700603 with colistin (MIC 0.25 μg/mL) as a positive control.

d Pseudomonas aeruginosa ATCC 27853 with colistin (MIC 0.25 μg/mL).

e Acinetobacter baumannii ATCC 19606 with colistin (MIC 0.25 μg/mL) as a positive control.

f Candida albicans ATCC 90028 with fluconazole (MIC 0.125 μg/mL) as a positive control.

g Cryptococcus neoformans ATCC 208821 with fluconazole (MIC 8 μg/mL) as a positive control.

Compounds 2–8 were evaluated for their anti-mycobacterial activity against bioluminescent derivatives of M. abscessus and M. marinum at a maximum concentration of 64 µg/mL (Fig. 5). Terrein (1) was not tested as we did not have enough sample. None of the compounds reached the threshold of a 1 log reduction in light output and MBCs were above the tested maximum of 1 mg/mL.

Figure 5 Activity of pure compounds against M. abscessus BSG301 and M. marinum BSG101.

Key: Compounds were tested at a concentration of 64 µg/mL. Data are presented as the log reduction in area under curve (AUC) values derived from the luminescence output over 24 h when compared to the DMSO control. The dotted line denotes a 90% reduction in AUC compared to the DMSO control and is used to denote the minimum inhibitory concentration. Raw data is available at: https://doi.org/10.17608/k6.auckland.20126423.v1.

Discussion

Secondary metabolites from A. terreus including terrein (1), aspulvinone E (2), flavipesolide C (7), butyrolactone II (8), and butyrolactone I (9) have been reported to exhibit a variety of bioactivities including anti-proliferative, anti-inflammatory, and antioxidant activities (Haritakun et al., 2010; Wang et al., 2011; Liao et al., 2012; Zhang et al., 2015; Guo et al., 2016; da Silva et al., 2017; Sun et al., 2018; Asfour et al., 2019; Tilvi et al., 2021; Buachan et al., 2021; Ghfar et al., 2021). The antimicrobial activities of terrein (1), terretonin (4), terretonin A (5), butyrolactone II (8) and butyrolactone I (9) have also been previously investigated; 4 (Fukuda et al., 2014; Salendra et al., 2021) and 5 (Li et al., 2019) were not found to be active. There have been several contrasting reports with respect to the antimicrobial activities of 1, 8 and 9 over the years. One study reported that both 1 and 9 displayed no antimicrobial activity against Bacillus subtillis, Candida albicans, Chlorella vulgaris, Chlorella sorokiniana, E. coli, Mucor miehi, Pythium ultimum, Rhizoctonia solani, Scenedesmus subspicatus, S. aureus, and Streptomyces viridochromogenes in disk diffusion assays (Nagia et al., 2012). This agreed with other studies which reported that both 1 and 8 exhibited no activity against A. baumannii, Enterococcus faecalis, E. coli, K. pneumoniae, S. aureus, and MRSA in disk diffusion assays (Qi et al., 2021) with 8 also exhibiting no activity (MIC > 100 µg/mL) against Enterococcus faecalis, K. pneumoniae, S. aureus, MRSA, and methicillin-resistant S. epidermidis along with 9 in broth dilution assays (Peng et al., 2022). Our data supports these findings. Our data also shows that aspulvinone E (2) and aspulvinone G (3) display no activity against a range of bacterial and fungal species.

In contrast, other studies reported antimicrobial activity for terrein (1) against B. subtillis (MIC 50 µg/mL), E. coli (MIC 50 µg/mL), and S. aureus (MIC 50 µg/mL) (Malmstrøm et al., 2002), as well as against Aeromonas hydrophila (IC50 > 20 µg/mL) and Enterococcus faecalis (IC50 20 µg/mL) (Goutam et al., 2017, 2020). Another more recent study found activity against Alternaria brassicae (MIC 8 µg/mL), Edwardsiella tarda (MIC 16 µg/mL), E. coli (MIC 4 µg/mL), Physalospora piricola (MIC 16 µg/mL), and S. aureus (MIC 32 µg/mL) (Li et al., 2020) for 1. Terrein has also been investigated for antimicrobial activity against Enterobacter aerogenes (MIC > 100 µM) and P. aeruginosa (MIC >100 µM) (Wang et al., 2011) and was not found to be active. However, it has been shown to inhibit quorum sensing and 3,5-cyclic diguanylic acid (c-di-GMP) sensing in P. aeruginosa, reducing virulence factor expression and biofilm formation with no effect on cell growth (Kim et al., 2018).

The antimycobacterial activity of 8 and 9 has also been investigated. While 9 was found to exhibit potent activity against the protein tyrosine phosphatase B of Mycobacterium tuberculosis (IC50 5.11 ± 0.53 µM) (Luo et al., 2019), both natural products were found to be inactive against M. tuberculosis (MIC > 50 µg/mL) (Haritakun et al., 2010). Our data supports these findings and indicates that the lack of activity of 8 extends to other mycobacterial species.

Of most relevance to our finding of anti-C. neoformans activity for terrein (1), this compound has previously been shown to exhibit activity against a range of fungal plant pathogens, including Alternaria solani (IC50 < 0.1 µg/mL) the causative agent of early blight in tomato and potato plants, Fusarium graminearum (IC50 5.7 µg/mL) the causative agent of head blight in wheat and barley, F. oxysporum f. sp. momordicae (IC50 1.7 µg/mL) a causative agent of fusarium wilt in bitter gourd, and F. oxysporum f. sp. cucumerinum (IC50 0.8 µg/mL) a causative agent of vascular wilt in cucumber plants (Gressler et al., 2015; Lu et al., 2017).

To the best of our knowledge, this is the first report of activity for terrein (1) against the opportunistic human pathogen Cryptococcus neoformans. C. neoformans and the related C. gattii are the major species responsible for life-threatening cryptococcal meningitis, of which there are an estimated 223,100 cases globally each year, leading to 181,100 deaths (Rajasingham et al., 2017). During the pandemic, C. neoformans has also emerged as a cause of opportunistic infection in some COVID-19 patients (Abdoli, Falahi & Kenarkoohi, 2021; Chastain et al., 2022). Treatment options for cryptococcal infections are limited, with Cryptococcal species using a wide range of intrinsic and adaptive mechanisms to resist treatment (Iyer et al., 2021). While terrein is unlikely to make an effective treatment for C. neoformans, understanding how it exerts its antifungal activity may provide further avenues for drug development.

Conclusions

Investigation of A. terreus, ICMP 477, led to the isolation of nine known natural products which were identified as terrein (1), aspulvinone E (2), aspulvinone G (3), terretonin (4), terretonin A (5), asperteretal B (6), flavipesolide C (7), butyrolactone II (8) and butyrolactone I (9) upon structure elucidation. One of the limitations of our study is that we had limited amounts of the pure compounds and so were not able to test all compounds against all our target bacterial and fungal species. However, evaluation of the biological activities of a subset of these natural products against a wide range of bacterial and fungal human pathogens showed no activity for any of the tested compound except for terrein (1) which exhibited potent antifungal activity against C. neoformans. While no new natural products were discovered during this project, several known antimicrobial compounds were identified. Transposition of our current methods into a higher throughput screening method will enable the rapid study of far more promising fungal isolates, increasing our chances of identifying novel antimicrobial compounds.

We would like to thank Michael Schmitz and Tony Chen for their assistance with the NMR and mass spectrometric data and the Community for Antimicrobial Drug Discovery (CO-ADD), funded by the Wellcome Trust (UK) and The University of Queensland (Australia), for carrying out some of the activity and toxicity testing of the pure compounds.

Additional Information and Declarations

Competing Interests

Author Contributions

Data Availability

Siouxsie Wiles is an Academic Editor for PeerJ.

Melissa Cadelis conceived and designed the experiments, performed the experiments, analyzed the data, prepared figures and/or tables, authored or reviewed drafts of the article, and approved the final draft.

Alex Grey performed the experiments, analyzed the data, prepared figures and/or tables, authored or reviewed drafts of the article, and approved the final draft.

Shara van de Pas performed the experiments, analyzed the data, prepared figures and/or tables, authored or reviewed drafts of the article, and approved the final draft.

Soeren Geese performed the experiments, analyzed the data, authored or reviewed drafts of the article, and approved the final draft.

Bevan S. Weir conceived and designed the experiments, authored or reviewed drafts of the article, and approved the final draft.

Brent Copp conceived and designed the experiments, analyzed the data, authored or reviewed drafts of the article, and approved the final draft.

Siouxsie Wiles conceived and designed the experiments, analyzed the data, prepared figures and/or tables, authored or reviewed drafts of the article, and approved the final draft.

The following information was supplied regarding data availability:

The data is available at Figshare:

Wiles, Siouxsie; Geese, Soeren (2022): The activity of Aspergillus terreus ICMP 477 against antibiotic-sensitive and resistant strains of E. coli, K. pneumoniae, and S. aureus by zone of inhibition assay. The University of Auckland. Dataset. https://doi.org/10.17608/k6.auckland.20760364.v1.

Wiles, Siouxsie (2022): Activity of the pure compounds Asperteretal B, Aspulvinone E, Aspulvinone G, Butyrolactone I, Butyrolactone II, Flavipesolide C, Terretonin, and Terretonin A isolated from the fungus Aspergillus terreus ICMP 477 against Mycobacterium abscessus and M. marinum. The University of Auckland. Dataset. https://doi.org/10.17608/k6.auckland.20126423.v1.

Wiles, Siouxsie (2022): Antibacterial activity of extracts from Aspergillus terreus ICMP 477 against Escherichia coli and Staphylococcus aureus. The University of Auckland. Dataset. https://doi.org/10.17608/k6.auckland.20113922.v1.

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
