# Peer review of "Terrien, a metabolite made by Aspergillus terreus, has activity against Cryptococcus neoformans"

_PeerJ, doi:10.7717/peerj.14239_

## Round 0.1 · original submission · Major Revisions

Three referees have reviewed your manuscript, and their comments are attached below.

Reviewer 1 ·

Basic reporting

1.The author should have a figure that showed the anti-Cryptococcus neoformans form terrien(1), as it's the only one purified compound that effective.

Experimental design

1. What's the criteria to choose the microorganism tested for antimicrobial assay?
Why the mold fungi did not test for antimicrobial assay, as data form the literature reported that terrien (1) showed antimicrobial activity against mold.
2.The author defined the MIC as causing a 1-log reduction in light production, what about the MBC/MFC?

Validity of the findings

1. Even the purified compounds are known chemical structure, but the new finding of the microbial activity was interesting with future study of mechanism.
2. If possible, the antimicrobial activity test with combination of two compounds will show more effective against microorganism than single compound.

Additional comments

Even the purified compounds are known chemical structure, but the new finding of the microbial activity was interested with future study of mechanism.

Annotated reviews are not available for download in order to protect the identity of reviewers who chose to remain anonymous.

Reviewer 2 ·

Basic reporting

(1) Introduction needs improvement. Please explain clearly to show how big the problem is and the significance of this study.
(2) There are duplications in writing.

Experimental design

(1) The investigation is quite rigorous, but not comprehensive.
(2) The methods were sufficiently described, except for the initial screening. Please explain the methodology of the initial screening.
(3) Is it not quite systematic. There are some gaps in the research flow that can raise questions from the reader because there is no explanation or reasoning. Please see the attachment which I have described point by point.
(4) Some results are missing.

Validity of the findings

no comment

Additional comments

Cadelis M, et al. manuscript “Terrien, a metabolite made by Aspergillus terreus, has activity against Cryptococcus neoformans” is an original article to explore the potential or novel compounds from the metabolites of microbial collected from New Zealand and the South Pacific regions. It is an interesting article. However, it needs improvement to make it comprehensively described. Some matters are not clearly explained or discussed.

Annotated reviews are not available for download in order to protect the identity of reviewers who chose to remain anonymous.

Reviewer 3 ·

Basic reporting

Its is a short manuscript and supplemental results should also be stated in the main section.

Experimental design

Design is confusing. Particularly, what is the rationale to use E. coli and S. aureus in the primary screen? What was the rationale of pursuing fractions 3 and 4 if fractions and 5 showed activity against S. aureus. Selected compounds were then tested against a bigger panel; what if other fractions had compounds with activities against other microorganisms as well, that were missed since the initial screen was performed in E. coli and S. aureus?

Validity of the findings

Conclusion is OK for what was tested.

---

## Round 0.2 · accepted · Accept

The authors have appropriately addressed the reviewers' concerns. The revised manuscript is now suitable for publication.

Reviewer 1 ·

Basic reporting

All questions are clear and accepted.

Experimental design

All method already done.

Validity of the findings

The results were discussed clearly.

Reviewer 2 ·

Basic reporting

No comment

Experimental design

No comment

Validity of the findings

No comment